# Spontaneous ssDNA stretching on graphene and hexagonal boron nitride in plane heterostructures

Binquan Luan[1]* & Ruhong Zhou [1]

Single-stranded DNA (ssDNA) molecules in solution typically form coiled structures, therefore stretching ssDNA is extremely crucial before applying any nanotechnology for ssDNA analysis. Recent advances in material fabrication enable the deployment of nano-channels to manipulate, stretch, sort and map double-stranded DNA (dsDNA) molecules, however nanochannels fail to stretch ssDNA molecules due to the ultra-short persistence length and the potential nonspecific-interaction-induced clogging. Given the significance of ssDNA stretching in genome analysis, here we report an ssDNA stretching platform: two dimensional in-plane heterostructure comprising graphene and hexagonal boron nitride (h-BN), and show that ssDNA can be stretched on a h-BN nanostripe sandwiched between two adjacent graphene domains ("nanochannel"). We further show that with a biasing voltage the stretched ssDNA can be electrophoretically transported along the "nanochannel", allowing easy controls/manipulations. When being conveniently integrated with existing atomic resolution sensors, the heterostructure platform paves the way for sequencing DNA on a planar surface.

[1] Computational Biological Center, IBM Thomas J. Watson Research, Yorktown Heights, NY 10598, USA. *email: bluan@us.ibm.com

Stretching ssDNA in solution has been predominantly studied in the past for its mechanical properties[1] using optical tweezers or atomic force microscopy (AFM). Amazingly, in a fully stretched ssDNA the spacing $a$ between the neighboring nucleotides can be about 2.1 times of that (~3.4 Å) in the canonical B-form dsDNA[2]. However, for the purpose of ssDNA sequencing with existing sensing technologies, such as AFM[3] and scanning tunneling microscope (STM)[4–6], it requires to stretch ssDNA on a planar surface which is extremely challenging given the nonspecific interaction (or sticking) between ssDNA and the surface. For example, limited studies have demonstrated that ssDNA molecules can be successfully (albeit with a low yield) adsorbed and stretched linearly by a flow on a modified mica surface[7], however, the relatively small averaged spacing ($a$ ~ 4.0 Å) indicates a partial stretching and potential intra-strand base stacking between the neighboring nucleotides. Therefore, a platform for stretching ssDNA with a great success rate is of vital importance and highly desirable in order to achieve the base sensing of ssDNA in an attoliter-scale sample[8].

Besides the mechanical stretching, it is straightforward to sterically stretch dsDNA confined inside solid-state nanochannels[9–13]. However, stretching ssDNA in a conventional nanochannel has remained impossible due to the electroosmotic flow[14] (as well as the entropy barrier) that prevents the entry of ssDNA. Even after its entry, the analysis of ssDNA in a nanochannel is challenging because of the fast electrically driven translocation[15]

and possible clogging. With recent advances in fabrication of two-dimensional (2D) nanomaterials, remarkably, different 2D nanomaterials can be integrated together to form an in-plane heterostructure with controlled domain sizes, including seamlessly connected graphene and h-BN[16], $WS_2$ and $MoS_2$[17], $MoS_2$ and $MoSe_2$[18] as well as $MoSe_2$ and $WSe_2$[19]. Consistent with the adsorption of ssDNA on a flat graphite surface in experiment[20], we have previously shown that biomolecules can be preferably adsorbed on a 2D nanomaterial due mainly to the strong van der Waals (vdW) interaction[21,22]. Here, with emerging in-plane heterostructures, we are motivated to design a nanochannel/nanostripe on the heterostructure surface to stretch an adsorbed ssDNA molecule sterically (or via an energy trap), as illustrated in Fig. 1a.

In order to demonstrate how 2D in-plane heterostructures can be utilized to stretch ssDNA, we carry out the proof-of-principle molecular dynamics simulations that have been widely applied to predict/confirm nanoscopic processes not yet possibly observed in experiment[23,24]. By employing sophisticated force fields for describing molecular interactions (see the "Methods" section), we investigate the dynamic movements of ssDNA on the graphene/h-BN heterostructures using our high performance computing (HPC) clusters. Our results demonstrate that an ssDNA molecule can be efficiently stretched in a linear conformation on the 2D heterostructure, which is excellent for ssDNA analysis by atomic resolution sensors.

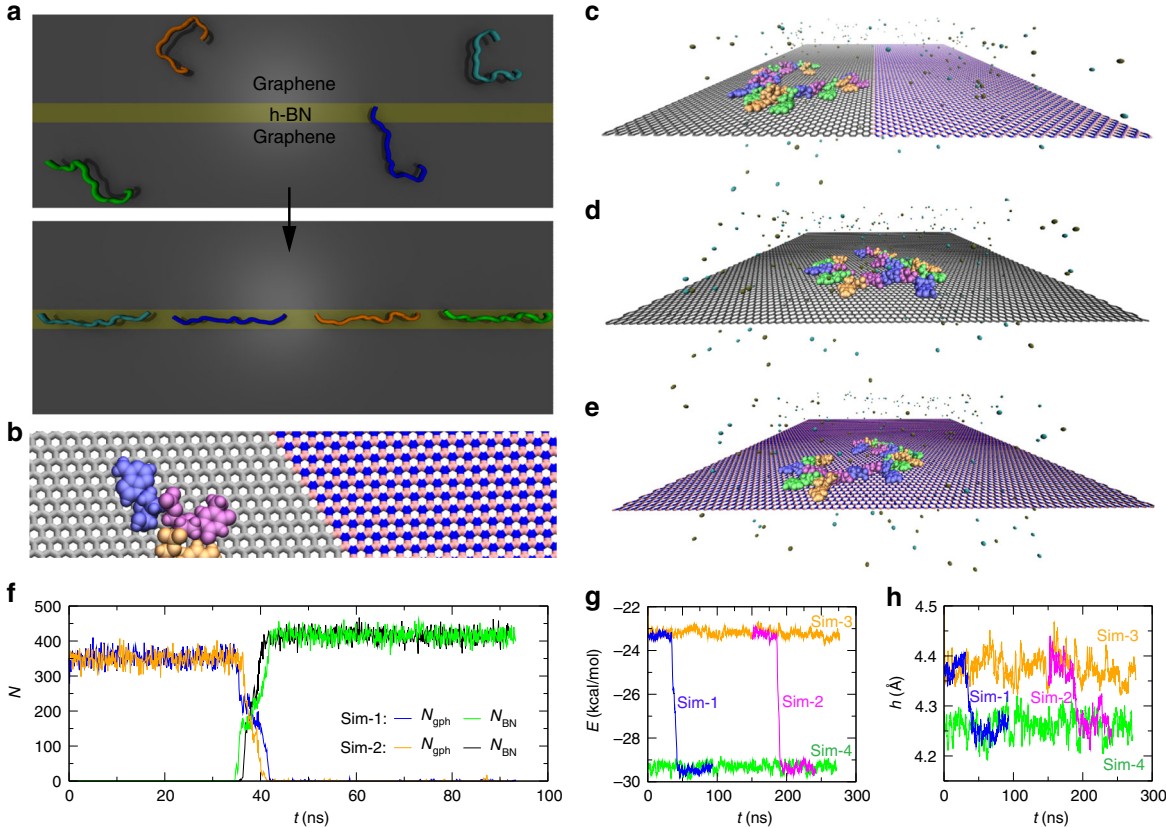

**Fig. 1** ssDNA dynamics on 2D materials: graphene, h-BN, and their in-plane heterostructure. **a** An illustration on how ssDNA can be spontaneously stretched on an in-plane graphene/h-BN/graphene heterostructure. **b** An enlarged view of an ssDNA fragment near the graphene/h-BN boundary, with atomic details. Carbon atoms in graphene are colored in gray while boron and nitrogen atoms in h-BN are colored in pink and blue, respectively. **c–e** Simulation systems of ssDNA on the graphene/h-BN heterostructure, graphene, and h-BN, respectively. A 20-base-long ssDNA is colored according to base types: A (blue), T (purple), C (orange), and G (green). Water is not shown; $K^+$ (tan) and $Cl^-$ (cyan) ions are shown as van der Waals spheres. 2D materials are in the stick representation. **f** Number of atoms in the graphene/h-BN heterostructure that are within 3.5 Å of ssDNA, during simulations. **g** van der Waals interaction energies (per nucleotide) between ssDNA and different 2D materials. Results from Sim-2 were shifted by 150 ns for clarity. **h** Heights of ssDNA (centers of mass) above different 2D materials

## Results

**ssDNA spontaneously diffuses from graphene to h-BN surfaces.** Firstly, we focus on the ssDNA dynamics near the boundary/junction of a graphene/h-BN heterostructure, as shown in Fig. 1b, c (also see Supplementary Fig. 1 in Supplementary Information for the entire simulation system). Before the ssDNA molecule reached the boundary, it diffused freely on the graphene surface with all nucleotide bases forming the strong $\pi$–$\pi$ interaction with graphene after being adsorbed from water (Supplementary Fig. 4), which is consistent with previous experimental studies[25]. Even though a complete modeling of the $\pi$–$\pi$ stacking would require an accurate description of the London dispersion (caused by favorable instantaneous-multipole/induced-multipole charge fluctuations) with high level quantum methods, modern force fields used in our simulations were found to be able to capture well the interaction strengths through fitting to experimental data[26]. Stochastically, one ssDNA nucleotide diffused across the boundary and arrived at h-BN, which is strikingly followed by the remaining nucleotides in ssDNA. Subsequently, ssDNA diffused freely on h-BN and never moved back to graphene. As shown in Fig. 1f, after about 35 ns, the number $N_{gph}$ of carbon atoms in graphene that were in contact with ssDNA decreased from about 350 to zero once ssDNA moved across the boundary, meanwhile the number $N_{BN}$ of boron and nitrogen atoms in h-BN in contact with ssDNA increased from zero to a saturated value (~420). This unidirectional motion of ssDNA was confirmed in two independent simulations (Sim-1 and Sim-2).

To unveil the transport mechanism, we carried out two control studies, where ssDNA moved on the graphene-only (Sim-3 and Fig. 1d) and on the h-BN-only (Sim-4 and Fig. 1e) 2D surfaces. Despite similar ssDNA dynamics and conformation sampling (see below) on graphene and h-BN, the adhesive interaction energies (including predominantly the vdW energies due to the charge neutrality of graphene or h-BN, Supplementary Fig. 2e) between ssDNA and these 2D materials can vary significantly. Figure 1g shows that the average interaction energy between ssDNA and graphene is about −23.2 kcal/mol per nucleotide (Sim-3), which is reduced by another 6.1 kcal/mol per nucleotide for interaction between ssDNA and h-BN (Sim-4), suggesting that ssDNA interacts with h-BN more strongly (yielding the driving force for ssDNA's unidirectional motion). Notably, this phenomenon qualitatively agrees with the adsorption energy difference of a small molecule on the graphene and on the h-BN discovered in previous experiment[27]. Consistently, interaction energies from Sim-1 and Sim-2 become more negative (i.e., more favorable) during the boundary-crossing events (Fig. 1g). It is worth emphasizing that ssDNA on graphene/h-BN can diffuse very fast (see Supplementary Fig. 2) despite the strong vdW adhesion, which allows ssDNA to reach graphene/h-BN boundary quickly through diffusion. The larger contacting number $N_{BN}$, as well as more favorable interaction energy $E$, are in agreement with the more intimate/closer contact between ssDNA and h-BN (Fig. 1h), as evidenced by the 0.12-Å-in-height reduction of ssDNA after diffusing from graphene to h-BN.

**ssDNA is stretched on the h-BN nanostripe sandwiched by two graphene domains.** With the above boundary-crossing phenomenon, we further studied the possibility of confining ssDNA on a graphene/h-BN/graphene heterostructure[16]. By limiting the width of the h-BN domain to be about 1.8 nm, the 2D heterostructure becomes a h-BN nanostripe sandwiched between two graphene domains. Effectively, such an in-plane heterostructure yields a surface channel for ssDNA with a width of 1.8 nm and a depth of 0.12 Å, reminiscent of much larger nanochannels for confining and stretching dsDNA. We performed two independent

simulations each for ssDNA with a linear (Sim-5 and Sim-6, Fig. 2a) and a circular (Sim-7 and Sim-8, Fig. 2b) conformations initially rested on graphene. The dynamics of ssDNA adsorption from water onto this heterostructure surface is discussed in Supplementary Information (Fig. S5). In Sim-5, we observed the anchoring of ssDNA on the h-BN stripe after diffusing towards it. Due to its flexibility, ssDNA quickly lost its initial linear conformation on graphene and folded into a hairpin-like structure with intra-strand base pairings (e.g. two A–T pairs as shown in Fig. 2c). For such "folded" conformation on the stripe, a fragment of ssDNA was left on the graphene surface (Fig. 2c). After about another 130 ns, thermal fluctuation and interaction energy optimization further drove ssDNA into a fully stretched conformation on the h-BN stripe (Fig. 2d, and Movie S1 in Supplementary Information). The ssDNA molecule remained in the stretched conformation because it was sterically confined and energetically trapped on the h-BN stripe (see Supplementary Fig. 3).

To highlight this driven process where the entropy energy of ssDNA on graphene was dominated by the enhanced adsorption energy of ssDNA on the h-BN stripe, in Fig. 2e, we show the scatter plot for numbers of graphene and h-BN atoms in contact with ssDNA. During the ssDNA stretching process, $N_{gph}$ and $N_{BN}$ decreased and increased respectively, with a gradual increase of interaction energy (indicated by colors in Fig. 2e).

Additionally, we analyzed the end-to-end distance $L$ of the flexible ssDNA during the stretching process. In Sim-5, due to the intermediate trapped and bent conformation, $L$ first decreased to about 35 Å but subsequently increased to 96.8 Å after stretching. In Sim-6, the initial linear conformation of ssDNA was preserved on graphene and was further stretched on the h-BN stripe (Fig. 2f). In Sim-7 and Sim-8, values of $L$ were small at the beginning (due to the initial circular conformation of ssDNA) and increased to 96.8 Å after ssDNA's stretching on the h-BN stripe (Fig. 2f). All four simulations unanimously suggested that ssDNA can diffuse towards and subsequently being stretched on the h-BN stripe.

From Sim-3 (Fig. 1c) and Sim-4 (Fig. 1d), we can theoretically calculate the persistence length $l_p$ (over which correlations in the tangent direction of a polymer are lost) of ssDNA on graphene and h-BN surfaces. According to the widely used worm-like chain (WLC) model, the correlation between two unit tangent vectors $\vec{x}$ along ssDNA ($s$) decays exponentially with their (contour) separation distance $Na$, i.e. $\vec{x}(s) \cdot \vec{x}(s + Na) = \cos(\theta) = e^{-Na/l_p}$, where $a$ (=6.25 Å) is the mean distance between the neighboring nucleotides on the 2D surface and $N$ is an integer. The inset of Fig. 2f shows that derived values of $l_p$ (from slopes) for ssDNA on both graphene and h-BN are comparable (~1.39 nm).

Theoretically, for a polymer chain confined in a nanochannel/nanostripe (with the width $w$ ~ 1.8 nm), the case when $l_p < w < 2l_p$ is categorized in the transition between the Gauss–de Gennes and the Odjik regimes[28]. With the known contour length $l_c$, the predicated end-to-end distance from the Gauss–de Gennes theory is $l_c l_p/w$ (~96.5 Å) and from the Odjik theory is $l_c[1 - A(l_p/w)^{-2/3}]$ (~99.8 Å, where the universal prefactor $A = 0.17$[29]), agreeing with our current simulation result of 96.8 Å.

**ssDNA molecules can be electrophoretically transported along the h-BN nanostripe.** Similar to the transport of dsDNA in a nanochannel, we further demonstrate that the graphene/h-BN/graphene heterostructure can be utilized to transport stretched ssDNA in an external biasing voltage $V$ (or alternatively by a shear flow). Starting from a stretched conformation of ssDNA on the h-BN stripe, we applied various electric fields ($V/L_x$, where $L_x$ is the length of the stripe) to electrophoretically drive ssDNA. As shown in Fig. 3a, when $V = 0.2$ V, ssDNA moved at a nearly constant velocity opposite to the field direction (see Movie S2

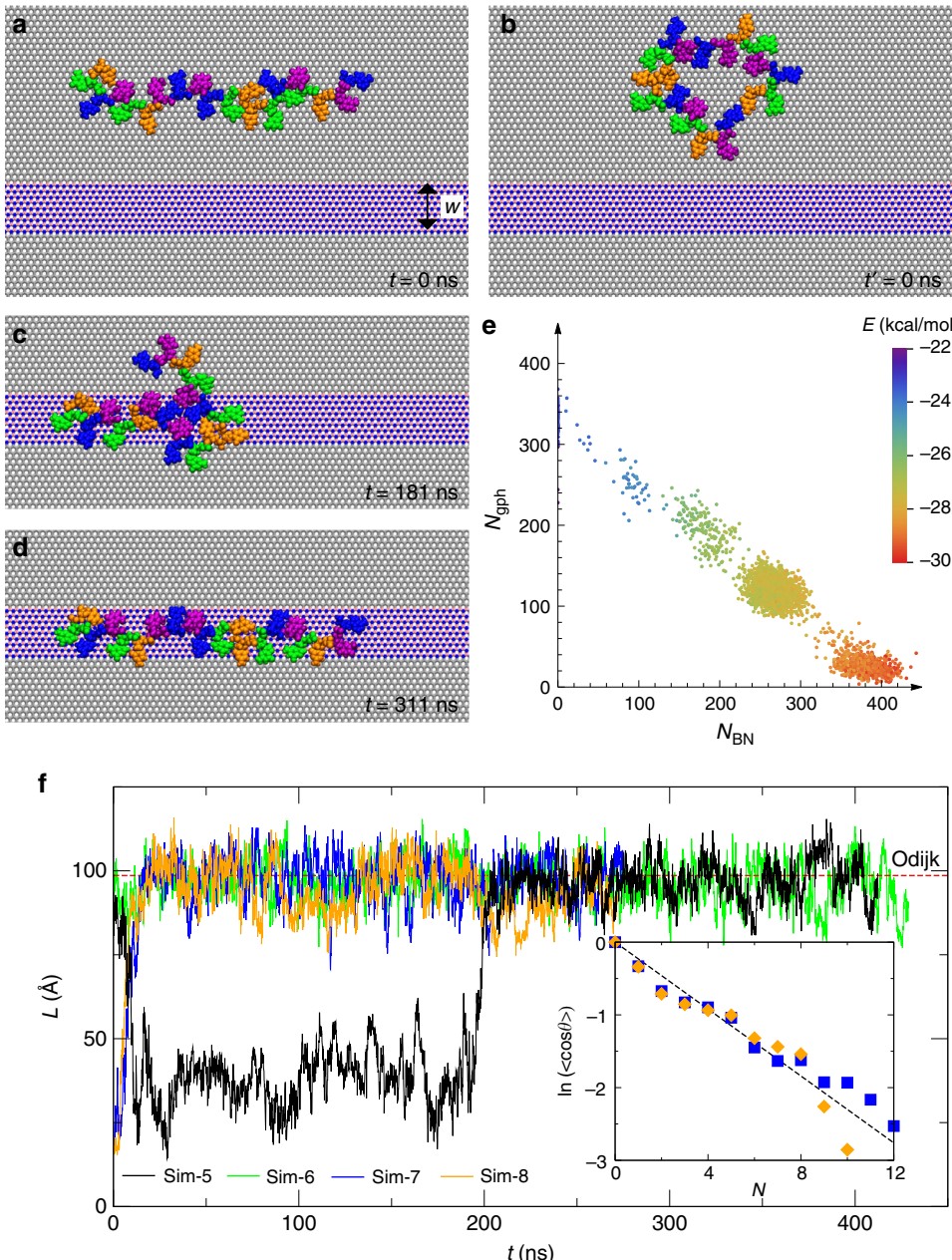

**Fig. 2** Modeling ssDNA motion on the graphene/h-BN/graphene heterostructure. **a** The initial linear conformation of ssDNA on the graphene domain (Sim-5 and Sim-6). **b** The initial circular conformation of ssDNA on the graphene domain (Sim-7 and Sim-8). **c**, **d** Conformations of ssDNA at 181 and 311 ns, respectively (Sim-5). **e** A scatter plot of numbers of atoms in the heterostructure that are in contact with ssDNA. The color shows the van der Waals interaction energy per nucleotide. **f** End-to-end distance for ssDNA during all four simulations. *In-set*: exponential decay of the correlations between two unit tangential vectors along the ssDNA backbone and separated by $N$ nucleotides in-between. Results were extracted from simulations of ssDNA on graphene only (Sim-3, blue squares) or h-BN only (Sim-4, orange diamonds)

in Supplementary Information). The displacement $X$ of ssDNA in various biasing voltages are illustrated in Fig. 3b. From these data, we obtained a linear relation between electrophoretic velocities $v$ of ssDNA and the applied biasing voltages (Fig. 3c), which yields the electrophoretic mobility of ssDNA $\mu = vL_x/V = 51.6$ nm²/ns V. When $V = 0$ V, ssDNA diffused back and forth. From the recorded position data $X(t)$ (Fig. 3b), we calculated the mean square displacement $\langle \Delta X^2 \rangle$ as a function of a time interval $\Delta t$ (Fig. 3d) and extracted the 1D diffusion constant that $D = \langle \Delta X^2 \rangle / 2\Delta t = 0.13$ nm²/ns. Therefore, with $k_B$ the Boltzmann constant and $T$ the temperature, the effective charge $q_{eff}$ of the stretched ssDNA confined on the h-BN stripe is $k_B T\mu/D \sim 10.2e$, which

indicates that about 46% of the ssDNA charge (compared with 12% from the Manning counterion condensation theory[30]) was screened, highlighting the complex interplay of electrostatic and hydrodynamic screening of ssDNA on the h-BN stripe. Therefore, similar to dsDNA in a conventional nanochannel, the motion of stretched ssDNA can be further controlled or manipulated for various applications (such as sorting and sequencing). It is noteworthy that the sensing of ssDNA bases through electron tunneling currents above a 2D surface and the transport of ssDNA in an electric field can be two independent steps, thus avoiding possible mismatch of the sensing and the transport speeds.

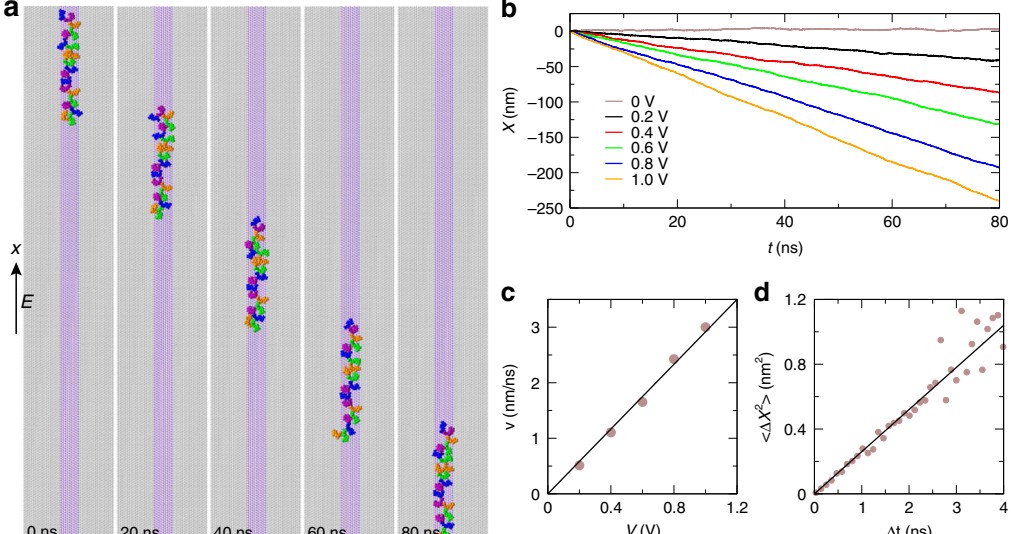

**Fig. 3** Electrophoretic transport of ssDNA on the graphene/h-BN/graphene heterostructure. **a** Illustration of electrically driven motion of ssDNA at different times ($V = 0.2$ V). The electric field is applied along the h-BN stripe. **b** Time-dependent displacements of ssDNA in various external biasing voltages. **c** Mean velocities of ssDNA at various applied biasing voltages. **d** Mean-square-displacements of ssDNA at different time intervals, when $V = 0$ V

## Discussion

For conventional nanochannels, it is impossible for a long dsDNA molecule to spontaneously enter a small nanochannel opening due to the large entropy barrier, and typically requires a large biasing voltage or a pressure-driven flow for dsDNA's entry into the nanochannel. For the current heterostructure nanostripe (or planar surface nanochannel), however, ssDNA (with $N$ nucleotides) can spontaneously enter it through the channel's long edge because of the flatness of the heterostructure and the entropy barrier ($\sim N^{0.5}$) outweighed by the stronger interaction ($\sim N$) between ssDNA and h-BN. These scalings with the ssDNA length ($N$) suggest that it is beneficial for a long ssDNA to be stretched on the heterostructure (i.e. a good read length of ssDNA during sequencing). Another advantage is that all DNA bases lie parallelly with the surface (via the $\pi-\pi$ stacking), which reduces spatial fluctuations of DNA bases (that exist in other ssDNA-stretching method[7]) and thus can improve the signal-to-noise ratio for base sensing (i.e. reducing error rate).

In conclusion, we have suggested a planar surface nanochannel for ssDNA stretching on the patterned 2D graphene/h-BN/graphene heterostructure surface. The facts of fast diffusion of ssDNA on such surface and stronger interaction energy for ssDNA on the h-BN stripe permit the remarkable spontaneous stretching of ssDNA to a great extent ($a \sim 6.25$ Å). Inspired by this result, generally one can conclude that selecting a 2D material with its $\pi$-electron energy close to HOMO energy levels of DNA-bases is one of the factors for a successful elongation of ssDNA on the nanostripe. Further stretching is possible by decreasing the ion concentration, reducing the width of h-BN, or reducing the dielectric constant of environment. It is worth noting that ssDNA stretching may occur on the h-BN/graphene/h-BN heterostructure surface as well, if the surface of graphene nanoribbon (connected to a power source) becomes positively charged. Besides ssDNA, the same principle can be used to stretch protein molecules on the 2D heterostructure surface. Conceivably, our proposed ssDNA stretching platform is compatible with existing sensors, such as the atomic-resolution AFM[3], high-resolution AFM[31] and STM[4–6] and allows parallelization when multiple nanostripes are built in the heterostructure. Therefore, it is expected that our findings will benefit future studies in DNA[8] and even protein[32,33] sequencing.

## Methods

**Molecular dynamics simulations**. To model the dynamics of an ssDNA molecule on the 2D graphene/h-BN in-plane heterostructure, we carried out all-atom molecular dynamics simulations using the program NAMD[34] on the IBM Power system HPC clusters. Figure 1b illustrates the nanoscopic simulation system (labeled as Sim-1 and Sim-2): the heterostructure (that measures $17.7 \times 17.9$ nm$^2$) contains 50% graphene and 50% h-BN connected seamlessly (Fig. 1b). The entire heterostructure was further solvated with a 100 mM KCl electrolyte, containing 69,990 water molecules, 151K$^+$ and 132Cl$^-$. A 20-mer ssDNA molecule with a random sequence of ATCGTAGCCGATATCGTAGC was initially placed in the electrolyte above the heterostructure (Supplementary Fig. 4) and was later adsorbed on the graphene surface due to the strong $\pi-\pi$ stacking (Supplementary Fig. 4b, Fig. 1b–d). For comparisons, we modeled ssDNA motions on either the graphene nanosheet (labeled as Sim-3 shown in Fig. 1c) or the h-BN nanosheet (labeled as Sim-4 shown in Fig. 1d). During simulations (production runs), atoms in various (designed) 2D nanosheets were fixed at their initial lattice positions (lattice constant: 2.5 Å) while the adsorbed ssDNA moved freely. In all simulations, the periodic boundary condition was applied in all the three directions.

The graphene/h-BN/graphene heterostructure used in our simulation is constructed by placing a h-BN segment with its width reduced to 1.8 nm forming a 1D h-BN nanostripe sandwiched between two graphene segments. The ssDNA molecule was placed initially on the graphene surface, with a linear (labeled as Sim-5 and Sim-6 shown in Fig. 2a) or a circular (labeled as Sim-7 and Sim-8 shown in Fig. 2b) conformation. Additional simulations (see Supplementary Materials) were carried out to show the same stretching effect can be achieved if the ssDNA was initially placed in water (about 2 nm above the heterostructure).

We applied the CHARMM27 force field[35] for the ssDNA molecule, the TIP3P model[36] for water molecules and the standard force field[37] for the ions. The force fields for graphene and h-BN were adopted from our previous studies[22,38]. In all 2D nanosheets, carbon atoms in graphene are neutral, while boron and nitrogen atoms in h-BN are charged with $0.4e$ and $-0.4e$ ($e$: the elementary charge) respectively. In this study, each C atom forming a bond with a N atom has been assigned a charge of $+0.133e$, and each C atom forming a bond with a B atom a charge of $-0.133e$, to maintain the "local charge neutrality" (neutralizing edge N/B atoms of h-BN). As shown in Supplementary Fig. 2e, the electrostatic interaction is dominated by the vdW one and thus simulation results are not sensitive to these charge assignments.

The simulation time-step with rigid bonds applied was 2 fs; non-electric (such as vdW and bonded) and electric interactions were calculated every 2 and 4 fs, respectively. The pair-wise vdW interaction was calculated using a smooth (10–12 Å) cutoff. We used the particle-mesh Ewald (PME) method with a mesh size of about 1 Å to calculate the long-range Coulomb interaction. After the minimization/equilibration, production runs were carried out in the NPT ensemble. The pressure was kept at 1 bar using the Nosé–Hoover method[39]. The temperature was kept constant at 300 K by applying the Langevin thermostat[40].

## Data availability

Data supporting the findings of this study are available within this article (and its Supplementary Information file) and from the corresponding author upon reasonable request.

## Code availability

Computer codes supporting the findings of this study are available from the corresponding author upon reasonable request.

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

## Acknowledgements

We would like to thank Tien Huynh for a critical reading of the manuscript and Bruce Berne for helpful discussions. The authors gratefully acknowledge the financial support from the IBM Bluegene Science Program (Grant numbers: W1258591, W1464125, and W1464164).

## Author contributions

B.L. conceived the concept, designed MD simulations, and analyzed data; B.L. and R.Z. discussed and wrote the paper together.

## Competing interests

The authors declare no competing interests.
