## [Peer Review File · Nature Communications]

Editorial Note: This article was originally reviewed at Nature Nanotechnology before being transferred to Nature Communications.

Reviewers' comments:

Reviewer #1 (Remarks to the Author):

The authors fully improved the manuscript and addressed my questions satisfactorily. I think that the manuscript is ready for publication in Nature Communications.

Reviewer #2 (Remarks to the Author):

1. The authors tried to address Reviewer #2's first concern by arguing that their ssDNA stretching method may be useful in new-generation DNA sequencing methods, not the nanopore-based ones. They suggested that a potential candidate was the electron tunneling approach. However, persist challenges remain for this approach, regarding read length, error rate and parallelization. Meanwhile, I'm not aware of any other sequencing platforms that may be compatible with the proposed ssDNA stretching method, which requires a heterostructure nanostripe. Please add further discussions on this issue.

2. As for Reviewer #2's second concern, the use of classical force fields (please specify CHARMM version) is fine to me. However, it seems that all the 2D sheets have a constant lattice constant (2.5 Å) in graphene, h-BN and even graphene/h-BN interface, and that a random charge (+/-0.13e) is assigned to interfacial carbon atoms (not specified). The realistic structure and charge distribution in the heterostructure (especially at the graphene/h-BN interface) have to be taken into account. This issue is crucial for ssDNA diffusion on the 2D surface.

3. I have an additional concern on the ssDNA length. I'm wondering whether the present stretching method is applicable for a very long ssDNA, because achieving long reads is the basic demand for single molecule sequencing approaches.

Reviewer #1 (Remarks to the Author):

The authors fully improved the manuscript and addressed my questions satisfactorily. I think that the manuscript is ready for publication in Nature Communications.

Response: We thank the Reviewer for his/her very positive comments.

Reviewer #2 (Remarks to the Author):

1. The authors tried to address Reviewer #2's first concern by arguing that their ssDNA stretching method may be useful in new-generation DNA sequencing methods, not the nanopore-based ones. They suggested that a potential candidate was the electron tunneling approach. However, persist challenges remain for this approach, regarding read length, error rate and parallelization. Meanwhile, I'm not aware of any other sequencing platforms that may be compatible with the proposed ssDNA stretching method, which requires a heterostructure nanostripe. Please add further discussions on this issue.

Response: We added discussion regarding read length and parallelization (page 11). The error rate should be low given the existing high-resolution imaging tools, STM (ref. 8-10), AFM (ref. 7 and newly added ref. 31). We added a new ref. 31 (Science, 365, 142-145, 2019), published two weeks ago by our IBM colleagues in Zurich, that shows a high-resolution AFM can reveal the detailed molecular structure. We propose that similar high-resolution AFM like the one just published in Science might scan the ssDNA-adsorbed 2D surface to obtain the DNA sequence information. We do agree that a practical usage of this novel technique still takes time and effort. We have added some comment along these lines (see pages 11 and 12).

2. As for Reviewer #2's second concern, the use of classical force fields (please specify CHARMM version) is fine to me. However, it seems that all the 2D sheets have a constant lattice constant (2.5 Å) in graphene, h-BN and even graphene/h-BN interface, and that a random charge (+/-0.13e) is assigned to interfacial carbon atoms (not specified). The realistic structure and charge distribution in the heterostructure (especially at the graphene/h-BN interface) have to be taken into account. This issue is crucial for ssDNA diffusion on the 2D surface.

Response: Good questions. We have added the CHARMM force field version (CHARMM27). The graphene sheet can seamlessly connect with a h-BN sheet as shown in experiment (ref. 6). In this study, each C atom forming a bond with a N atom has been assigned a charge of +0.133e, and each C atom forming a bond with a B atom a charge of -0.133e, to maintain the "local charge neutrality" (neutralizing edge N/B atoms of h-BN). Both graphene and h-BN sheets belong to the so-called "van der Waals family" of 2D nanomaterials, i.e., their interactions with other biomolecules such as DNA are dominated by the van der Waals interactions. For example, as shown in Fig. S2e in supplementary information, even though both boron and nitrogen atoms in h-BN are charged (+/-0.40e), their electrostatic interactions with ssDNA are negligible. Similarly, even if the charges of C atoms on the graphene edge are not accurate, their electrostatic interactions with ssDNA will still be negligible as compared to their van der Waals contributions. Therefore, we believe our conclusions will not be sensitive to the charges assigned to these C atoms. We have added some comments on this (pages 13 and 14).

3. I have an additional concern on the ssDNA length. I'm wondering whether the present stretching method is applicable for a very long ssDNA, because achieving long reads is the basic demand for single molecule sequencing approaches.

Response: We emphasized that for a long ssDNA (length =N), the entropy barrier for ssDNA scale with the length as $\ln(N)$ but the potential energy decrease scales as N (see the highlighted text on page 11). Therefore, the reading length is not an issue.

REVIEWERS' COMMENTS:

Reviewer #2 (Remarks to the Author):

This manuscript can now be published in Nature Communications.